# High-Rise Building 3D Reconstruction with the Wrapped Interferometric Phase

**DOI:** 10.3390/s19061439

**Published:** 2019-03-23

**Authors:** Rui Guo, Fan Wang, Bo Zang, GuoBin Jing, Mengdao Xing

**Affiliations:** 1School of Automation, Northwestern Polytechnical Universtiy, Xi’an 710129, China; 2Nanjing Research Institute of Electronic Technology, Nanjing 210039, China; wsxyxh@163.com; 3School of Electronic Engineering, Xidian Universtiy, Xi’an 710071, China; bzang@mail.xidian.edu.cn (B.Z.); xmd@xidian.edu.cn (M.X.); 4Huawei Technologies Co. Ltd., Shenzhen 518129, China; guobinjing01@163.com

**Keywords:** high-rise building, wrapped InSAR phase, layover, 3D information

## Abstract

The great development of high-resolution SAR system gives more opportunities to observe building structures in detail, especially the advanced interferometric SAR (InSAR), which techniques attract more attention on exploiting useful information on urban infrastructures. Considering that the high-rise buildings in urban areas are quite common in big cities, it is of great importance to retrieve the three-dimension (3D) information of the urban high-rise buildings in urban remote sensing applications. In this paper, the 3D reconstruction of high-rise buildings using the wrapped InSAR phase image was studied, referring to the geometric modulation in very high resolution (VHR) SAR images, such as serious layover cause by high-rise buildings. Under the assumption of a rectangular shape, the high-rise buildings were detected and building façades were extracted based on the local frequency analysis of the layover fringe patterns. Then 3D information of buildings were finally extracted according to the detected façade geometry. Except for testing on a small urban area from the TanDEM-X data, the experiment carried on the single-pass InSAR wrapped phase in the wide urban scene, which was collected by the Chinese airborne N-SAR system, also demonstrated the possibility and applicability of the approach.

## 1. Introduction

With the remarkable improvements in radar sensor, decimeter very-high-resolution (VHR) synthetic aperture radar (SAR) made it possible to map urban areas in very high level of detail [1,2,3,4]. In particular, together with advanced interferometric SAR (InSAR) techniques, now it is possible to obtain information of a single structure from VHR SAR images [5,6,7,8,9]. With the data collected by modern high-resolution SAR sensors, including both the airborne and spaceborne systems, urban areas characterized by different types of objects can be analyzed in much more detail referring to the geometric modulation of SAR imagery such as shadow, foreshortening and layover [5,6]. With VHR SAR data, the individual building information can be well-understood by analyzing and studying useful information contained in SAR signatures e.g., layover [3,4]. In urban area, building information retrieval is an important task for SAR applications in many fields, such as city planning and disaster monitoring [2].

As a hot topic in urban remote sensing, building information retrieval and reconstruction, from SAR images, have been extensively investigated. Owing to the spatial resolution of up to 1 m compared to medium (10~30 m)- and high (3~10 m)-resolution SAR data, a single SAR image is used to detect buildings, to estimate building parameters, and to retrieve the isolated building height [3,10,11,12,13]. To conquer the shortages while using only one SAR amplitude image, multi-aspect SAR data containing several SAR amplitude images is utilized to extract and reconstruct buildings, based on the building projection model [14,15,16]. The quad-polarimetric characterization allowed building detection [16], and, images from other sensors, such as optical images were also introduced to assist the building of 3D information extraction [17,18,19]. However, using amplitude information alone can only provide limited information.

High-resolution InSAR technique has been demonstrated quite effective for building projection model descriptions as well as parameter inversion, which attracts more attention especially with the growing accessible VHR InSAR data recently. Combined with the advantages of very high resolution, building infrastructure information retrial with InSAR data keeps a hot topic in urban application. In 2000, Petit et al. [20] exploited the spectral shift between the interferometric image pair in order to separate the vertical signal from the horizontal signal, and Gamba et al. [21] proposed a method to extract and characterize the building structures from the 3D terrain elevation data, which utilized a modified machine vision approach and tested on several large buildings.

Benefit from the modern high-resolution spaceborne InSAR systems such as TanDEM-X, many researchers have studied further based on the single interferogram. As building layovers create a regular pattern in the mapping counter, Rossi et al. [3] utilized the behavior of displaced pixels within a cell on the geocoding stage in the interferometric processor, to extract the layover regions without the use of external supports. Thiele and Dubois et al. researched the former works of building reconstruction using multi-aspect InSAR data [15,22]. They have proposed several different building detectors, such as phase ramp detectors to detect medium- and high-rise buildings from their layovers [6,9,23,24] and extract building parameters, using parameterized model fitting.

Whereas developing similar InSAR urban remote sensing application in China is a quite challenge as high-rise buildings, higher than 24 m or, buildings with more than ten floors, are quite popular. Both the public buildings and residential buildings result in complex signatures of building projections in the InSAR image. Hence, some research focused on high-rise building analysis recently [7,8,25,26]. In VHR SAR images, the projection of high-rise building has a special characterization [7,19] compared to other building structures, and this issue has been developed further in much research, on this basis. In our former work [7], Guo and Zhu extracted the high-rise building features from amplitude and coherence data, which is demonstrated by using TanDEM-X data. Zhang et al. [8] proposed a non-local frequency estimation method to eliminate fringe patterns of high-rise building layover. Wen [25] and Tang [26] et al., detected high-rise buildings after analyzing the characterization of high-rise buildings in phase, and amplitude, respectively. For high-rise buildings, instead of an unwrapped phase from interferogram, the wrapped phase showed its power in high-rise building layover signature analysis [5,7,8].

As mentioned above, it is important to interpret, and to retrieve, high-rise building information in urban area city planning. However, there is few research papers focusing on high-rise buildings in the wide urban scene among the aforementioned papers. For this purpose, the multi-mode VHR SAR data are collected by the airborne N-SAR system [27] from China for urban remote sensing activities. N-SAR is designed to operate in X-band currently, with simultaneous single-pass interferometric capability, as two radar antenna mounts are designed to fix the planar array antennae under two sides of the aircraft and the physical baseline is about 3.25m. Flight experiments were carried on mainly in the middle of China, and different terrains types were collected in experiments, including forest, buildings, grassy earth, crops, rivers, and mountains [27]. Flight operations were performed at altitudes up to 6000 m above mean sea level. The off-nadir angle ranges from 25° to 70°.

In this paper, the work of 3D information reconstruction of high-rise buildings was introduced by using the wrapped InSAR phase image under the assumption of rectangular shapes, which presented a predominant layover area. In Section 2, the features of high-rise building layovers in InSAR signature were discussed, especially the projection model and the wrapped phase. Then the 3D information reconstruction approach was proposed and described in detail with an example wrapped phase of a small urban area from TanDEM-X in Section 3. In Section 4, the measured data from airborne N-SAR systems was introduced and experimented to validate the possibility and effectiveness of the proposed approach. Finally some conclusions were summarized and future work was put forward in Section 5.

## 2. High-Rise building signature

### 2.1. High-Rise Building Layover

As similar to the scattering model and projection geometry illustrated in Figure 1a, the high-rise building has a dominant scattering signal of the front wall in interferogram, and interferograms always have serious layovers, especially in high-rise building areas. We adopt the simple flat-roof building scattering model, in which the roof of the building is modeled with much less scattering compared to the edges and wall structures in side-looking VHR image. The layover of these high-rise buildings has an inherent spatial characteristic, which is dominated by the distance between adjacent windows and typical height between the floors of three meters, as the windows and the walls are constructed with different materials. Considering the high-rise building level, the building model can be simplified to the flat roof and rectangular shape with width w and height *h* viewed by a SAR sensor with the off-nadir look angle θ. For high-rise building, the height satisfies:(1)h>>w⋅tg(θ)

In the projection geometry, *b* is double-bounce reflection. *a*, *c,* and *d* are the single backscatter reflection of ground, front wall, and roof, respectively. The figure *e* represents the shadow. While, *L* contains the layover and double-bounce backscattering in the ground-projected image space, and
(2)L=h⋅ctg(θ)

In the amplitude image, *R* is the length of building layover in SAR image and
(3)R=L⋅sin(θ)

It is well-known that the large bright layover is an important signature of the high-rise building in the amplitude image.

### 2.2. Wrapped Interferometric Phase of Layover

Figure 1b illustrates an example of the wrapped InSAR phase of high-rise building areas [7] from TanDEM-X, and areas without infrastructures with noisy interferometric phase. In building layover, the phases of walls are dominant, which is the same as in SAR projection. And the periodical fringe patterns are apparent for high-rise building wrapped phase in their layovers. As known, the ambiguity height is constant, while InSAR system parameters are fixed. Hence, the period number of the building layover fringe pattern varies on the building façade height. Higher building may have more periodical fringe patterns in their layover. Meanwhile, the phase fringe is also related to the building orientation ϕ as shown in Figure 1a. And the orientation of the phase fringe pattern in high-rise building layover is related to the range direction under the side-looking imaging geometry of SAR sensor. From above, it is apparent that building layover patches can be extracted by estimating the local fringe frequency.

As the strong noise in the interferometric phase could cause serious errors, interferogram should firstly be filtered as the pre-processing. It has been proved that non-local method achieves good filtering result while well retaining structure details such as linear features and edges [28,29,30]. Hence, the non-local InSAR filter [28] is carried on the interferogram to reduce the phase noise.

## 3. Methodology

In this section, the approach of high-rise building 3D information retrieval was introduced in detail according to the analysis of interferometric phase signature of high-rise building layover in Section 2. The proposed approach consists of building detection, façade extraction, and 3D information reconstruction, as shown in Figure 2. Generated by a pair of single-pass InSAR images, the interferogram is firstly calculated and then filtered with non-local method to get the wrapped de-noised phase.

### 3.1. Building Detection

From the analysis of high-rise building projection in the wrapped phase imagery, it is known that the high-rise layover characteristic can be exploited to help detect the urban buildings. 

As pointed out in Section 2.2, the local fringe frequency estimation can help detect the high-rise building layovers. The types of approaches to measure the local frequency have been provided and used in the context of volcano or glacier [31,32]. Herein, the method proposed by Vasile et al. [31] has been extended to eliminate the building layovers [8]. It has been proved that it is possible to extract high-rise building layovers from the periodical fringe patterns in wrapped phase imagery [8]. Hence, in our work the local frequency in the range direction is estimated by the following equation:(4)$$f_{slant}=\frac{1}{2\pi }\cdot \arg\left[\sum_{\left(p,q\right)}N_{p,q}N_{p+1,q}\cdot \gamma (p+1,q)\gamma^{H}(p,q)\right],$$
(5)$$\mathrm{where}\;\gamma (p,q)=E\left\{s(k,l)\cdot s{\left(k-p,l-q\right)}^{H}\right\}$$
where E{⋅} is the mathematical expectation, H represents complex conjugate and (k,l) denotes the pixel coordinates. s(⋅) is the phase signal corresponding to a two-dimensional (2D) complex sine wave of the interferometric ϕ(⋅), viz. s(⋅)=exp(jϕ(⋅)). While γ is the autocorrelation between two pixels, which is computed by spatial averaging of the products $s(k,l)\cdot s{\left(k-p,l-q\right)}^{H}$ for all the possible delays (p,q) within the window used in the paper. *N* represents the number of the delayed phases. In the paper, the size of window for local frequency estimation is set as 13 × 13. The local frequency in Equations (4) and (5) is usually in the interval (−0.5, 0.5) due to the drastic changes in the decorrelation phase [8,31]. The values of the slant range frequencies are shown as Figure 3a, corresponding to the tested example area in Figure 1b. 

After that, a threshold should be set to remove the decorrelation areas in the estimated frequency result, so two classes including layover candidates and rests are segmented. According to the statistical distribution of the estimated local frequencies, an empirical threshold is defined. Figure 3c,d illustrate the statistical distribution of the estimation values in the layover area, and other area, respectively. Apparently, the local frequency values in the layover area are irregular and most are larger than zero. Whereas in non-layover area, the values are nearly distributed regularly viz the zero-centered symmetry. Figure 3b shows the statistical distribution of the local frequency value in Figure 3a, the distribution center is still at zero but in asymmetry distribution. Thus, *t* = 0.02 is set as the empirical threshold, because from this value the symmetry distribution is destroyed. Figure 4a illustrates the layover candidates and rests by thresholding, and the values of rests lower than 0.02 are set to be zeros while the layover candidates keep their values.

To further extract the high-rise layover patches from the segmented layover candidates, the *k-means* method is used to classify the segmentation result [6]. After performing *k-means* classification for *k* = [3:6], *k* = 6 is the ideal number of clusters to obtain the desired result, and then the initial high-rise building layover map is achieved. Figure 4b shows the initial layover map using *k*-means method. It can be observed that most noise-like rests are removed.

Since there are still some other types of buildings (low- or medium-buildings) left in the layover map as noise, the layover map was filtered by setting a neighborhood window. Considering both the spatial resolution and the general size of high-rise building, the size of the window is set as 7 × 3. If the number of the surrounding pixels in the neighborhood window is more half of the whole pixel number in the window, this pixel is retrained as the layover pixel. 

Finally, to eliminate the remaining noise and to separate the single building layover from each other that may be in one connected patch, the morphological opening and closing in both range and azimuth direction are operated. The empirical parameters of morphological operation are set manually according to the datasets, and in Figure 4 the rectangle windows in sizes of 5 × 15 and 21 × 3 are adopted. After all the aforementioned steps, the single high-rise building layover is detected and labelled, which is shown in Figure 4c. Figure 4d is the optical image of these five high-rise buildings from Google Earth. 

### 3.2. Façade Extraction

The building layovers mainly contain the building façade information as discussed in Section 2.1. Therefore, the building façade belonging to single high-rise building are extracted in this part. Due to the building orientation towards the sensor in side-looking mode, not all the façades (main and side façade) are visible. In most cases only the main façade can be recognized from the building layover. Therefore, the key task is to determine the number of façades in one detected building layover. If the façade number is two, then it goes to the step to extract both the main and side façade, otherwise only the main wall is visible and extracted. 

Firstly, based on the labelled high-rise building layovers, the two-dimensional (2D) fringe local frequencies and directions can be estimated from the wrapped phase using maximum-likelihood (ML) frequency estimator [32]. By maximizing the 2D discrete Fourier transform, the 2D frequencies $f_{x}$ and $f_{y}$ are calculated corresponding to the peaks of the signal. With respect to $f_{x}$ and $f_{y}$, the normal orientation of the local fringe is achieved as
(6)$$\phi =\tan^{{}_{-1}}(f_{y}/f_{x})$$

And then this orientation is taken as the building façade orientation. If the orientation is less than zero, an angle of π will be added to retrieve the building façade orientation. Figure 5a shows the orientation estimation result viz. the orientation map. It can be found that in the orientation map Building 2 and Building 3 have quite different estimated orientation values on the two sides of an obvious boundary, such as the zoomed-in image of Building 2 illustrated in Figure 5b. And it implies that the two façades of the high-rise building could be distinguished from each other.

Then, the orientation map was utilized to determine the building façade number [24] as the orientation of the fringe pattern between two façades changes greatly. As shown in the upper panel of Figure 5c, there are two different orientations in two area in the building layover model, and the two different orientations are marked with different color. If two areas with two different orientations exist in the building layover, it is determined that two façades of the building are visible as the upper figure in Figure 5b. To determine the number of visible façades, the statistical distribution of orientation values within each detected layover patch is given. If two orientation peaks are found, it is believed that two façade of the building are visible [24]. 

If two façades were visible in the building layover, the edge between two façades should be determined to separate the layover into two parts. As the two façades building layover model shown in Figure 5c, the building edge between two façades is in the orientation jump position, whereas the orientation value changed greatly, as illustrated in the bottom panel of Figure 5c. The bottom figure in Figure 5b shows the mean orientation values along each column, and the orientation jump position can be detected to achieve the edge between two façades under the rectangular shape assumption of building. For example, in the low panel of Figure 5b, the red dashed line indicates the position of the orientation jump, as the difference of the orientation values between two sides of the gap needs above 90°. Then depending on the edge position, the building layover patch is segmented into two façade parts belonging to the main wall and the side wall respectively.

### 3.3. 3D Information Reconstruction

Based on the extracted high-rise building façades, 3D information of these detected buildings can be retrieved. Three parameters are necessary for 3D information reconstruction, including building length, width, and height. Under the building model assumption of a perfect rectangular shape and vertical wall, each building layover corresponds to one parallelogram shape or two adjacent parallelograms, which represent the only existed main wall or two walls including both main and side wall, respectively. Figure 6 shows the real building layover in SAR image and the matched parallelograms models in the situation of both two walls, and single wall, respectively. In Figure 6, it is apparent that the building length and width are related to the edges of the parallelograms. The edges are the ones which are not parallel to the range direction. While, the building height concerns to the edge parallel to the range direction.

From above, after façade extraction of each building layover, firstly the building height can be approximated as h=R⋅Δr/cosθ, where *R* is the layover length, Δr is the range resolution, and θ is the off-nadir look angle, which is the same as shown in Figure 1a. For each building, no matter only one or two façades are extracted, the building height can always be estimated from the parallelogram shown in Figure 6a,b.

While going to the length and width, the complementary edges of the parallelograms, which are not parallel to the range direction are employed. For building length *l*, the complementary edge of the parallelogram, representing the main wall, is used. Notably, the width w could be estimated only if both façades are visible as shown in Figure 6a. For the high-rise building layover with two façades, the edge of the second parallelogram indicates that the side wall gives the building width w; for one-façade building, only the main wall contributes to the layover signature, as shown in Figure 6b, so the width is not estimable. Unlike the building height related to the off-nadir angle, the façade orientation and azimuth resolution contribute to estimate the building length and width [6,7,9]. 

## 4. Experiment on wide urban area

### 4.1. N-SAR Test Data and Pre-Processing

The urban dataset in Figure 7a was collected from the area between the city center and the suburbs of an intermediate city in China by the airborne N-SAR system [1] in side-looking mode. The resolution is 0.36m in range and 0.2m in azimuth. This urban scene coverage is around 1.5km × 1.3km (range × azimuth). Figure 7b shows the corresponding optical image (@ Google Earth) of this area, where all types of buildings (including low, medium, and high-rise buildings) are contained. Three high-rise areas have been marked in optical image. Herein, in the three areas viz. Area 1, Area 2 and Area 3, there are 18, 5, and 12 high-rise buildings, respectively.

In Figure 7a, the wrapped phase from airborne SAR is quite noisy, and it is necessary to filter the InSAR phase image before implementing the high-rise building information retrieval. After non-local filtering, the fringe pattern became much clearer, as in Figure 8. And then, the main work of the paper begins with this filtered wrapped InSAR phase. In Figure 8, the phases of these high-rise buildings sometimes overlap with each other, because usually, the high-rise buildings are densely concentrated in both residential community and commercial areas. Therefore, for some high-rise buildings, only part of its phase fringe patterns can be recognized due to the phase aliasing.

### 4.2. High-Rise Building Layover Detection

During the high-rise building detection step, the local range frequency estimation result using 13 × 13 window is achieved as Figure 9a. Due to Nyquist-Shannon criterion [31], the estimated frequency values in the range direction lie in (−0.5, 0.5). According to the statistical analysis of estimated frequencies in Section 3, a threshold equal to 0.02 is set to get the segmentation result as Figure 9b, in which two classes viz the layover candidates and rests are displayed. The layover candidates, whose estimation values are larger than the threshold, keep their values; the rest, whose values are smaller than the threshold, are set to zero. To extract the possible high-rise building layover from the layover candidates, *k*-means clustering with *k* = 6 is used to achieve the initial layover map as Figure 9c. Apparently the layovers are mainly concentrated in the high-rise area. And then a 7 × 3 window slides through the k-means filtering result, to eliminate the remaining noise-like layovers, and then the final labelled layover map is obtained by morphological opening and closing, as shown in Figure 9d. The building numbers are also labelled in Figure 9d, including the missed buildings, which have not been detected, and the missed building number is with a circle. Based on the extracted high-rise building layovers in Figure 9d, the orientations are calculated, as shown in Figure 9e, by using the local frequency ML estimator carried on the layover fringes. 

For details, an enlarged orientation map of the region around building 1-3 in Area 3 is illustrated as Figure 10a. And from the vision, Building 2 has two façades and the others have only one façade, which can also be observed from the figures of the mean orientation value along the column for Building 1-3 in Figure 10b. And in Figure 10b, it is apparent that the orientation jump (orientation difference larger than 90°) only exists in Building 2. Moreover, as one building has two visible façades at most, the second jump position around the red circle is taken to achieve the façade edge.

### 4.3. High-Rise Building 3D parameters

In the N-SAR data experiment, the proposed approach is carried on the whole area in Figure 4, and the parameters, such as the window size used in Section 3, are all manually fixed consistently within the whole scene in Figure 7. Note that high-rise building detection is essential, as the entire work is based on its result. However, due to the overlapped layovers of the adjacent high-rise buildings, not all the buildings can be correctly detected.

Table 1 shows the detailed result of the high-rise building detection and 3D reconstruction. Among all the three high-rise areas marked in Figure 7b, nearly 72% of the buildings are detected, which can also be observed from Figure 9d. In Area 2, only 1/5 of the buildings are not detected, as the buildings are not so densely distributed as in Area 1 and Area 3. Also, the high-rise buildings, showing a bad orientation towards the sensor or hidden by neighborhood buildings were not well extracted.

From the known parameters of the system, the assumed height of ambiguity is obtained as nearly 19 m from Equation $h_{a}=\lambda r\sin\theta /2B_{⊥}$, where λ, r
$B_{⊥}$ represent the wavelength, the slant range, and the baseline, respectively. The retrieved height values of these detected buildings are from 24.39 to 40.95 m, and these are all larger than the ambiguity height 19 m. As pointed out in Section 2, more periodic fringe patterns indicates a higher building, therefore the estimated height value depends on the fringe pattern in the building layover. The mean height of the detected buildings in Area 1 is 35.98 m, larger than 26.89 m in Area 2 and 28.64 m in Area 3. And this implies more periodic fringes in the building layover in Area 1, which is consistent with the fringe observations in Figure 7a. The building heights are a little underestimated as the mean heights of the residential area are 34~48 m in the three areas. Take 34 m as a reference, the mean height error is also illustrated in Table 1. 

Among the detected high-rise buildings, around 44% of them are extracted with the two façades. By orientation value distribution analysis in each building layover patch, 11 buildings in 35 are extracted with two façades, and the width of these 11 buildings can be retrieved. For the rest of the one-façade buildings, their width cannot be estimated. And the 3D information have been all retrieved for the two-façade building extracted. The mean length of the detected high-rise buildings are 19.68, 13.49, and 23.63 m in Area 1, Area 2 and Area 3, respectively. The estimated mean width of two-façade buildings in the three areas are 12.83, 7.56, and 17.46 m. It is apparent that both the estimated building lengths and widths are smaller than the general length and width of the residential buildings. The 3D information retrieved will be examined in more details, such as by comparing the reference GIS parameters, which is in now lack. Usually the 3D parameters are underestimated, due to the overlapped fringe patterns between adjacent layovers and the loss of fringe patterns during the processing.

Figure 11 shows the reconstructed 3D models of the high-rise buildings overlaid on the amplitude image of this urban scene. For building extracted with only one façade, the width is given with a constant value. For building extracted with both two façades, it can be observed that the length and width of most buildings are underestimated.

## 5. Conclusions

The development of building information retrieval, with VHR InSAR, is one important research subject in urban remote sensing. In the paper, the wrapped InSAR phase image of the single interferometric pair of VHR SAR data is exploited for retrieving the high-rise building information. And the proposed approach shows its possibility on wide scene without using external information. Unlike most related works, this paper deals with the high-rise building in wide scene and utilizes only the wrapped phase image.

The presented methodology is on the assumption of a rectangular shaped building model, and the parameter of this type of high-rise buildings can be recovered using the fringe pattern characteristics within the layovers in the wrapped phase image. A small urban phase image from the TanDEM-X data is firstly showed in the paper for high-rise building characteristics analyses and methodology descriptions in details. The building detection is developed by using the local frequency estimation in slant range, which has been further improved in order to extract the building layover patches. Then the building façades are determined by analyzing the building orientation values. Under the rectangular shape building hypothesis, the building parameters are estimated as corresponding to the façade parallelogram. The possibility and applicability of the proposed method is also demonstrated on the wide urban scene collected by single-pass airborne N-SAR system.

Generally, most of the high-rise buildings are detected and only few are visible with two façades. Thus, the influence of the sensor condition and the building distribution, such as the incidence angle and building orientations, will be the object of future work. In the workflow, the building detector plays an important role. The parameterization of the detectors should be considered in future research, especially in applying the methodology to wide urban scene for more different data-sets. Finally, the top-down approach will be refined to achieve a totally automatic and robust approach for quite dense urban area.

## Figures and Tables

**Figure 1 sensors-19-01439-f001:**
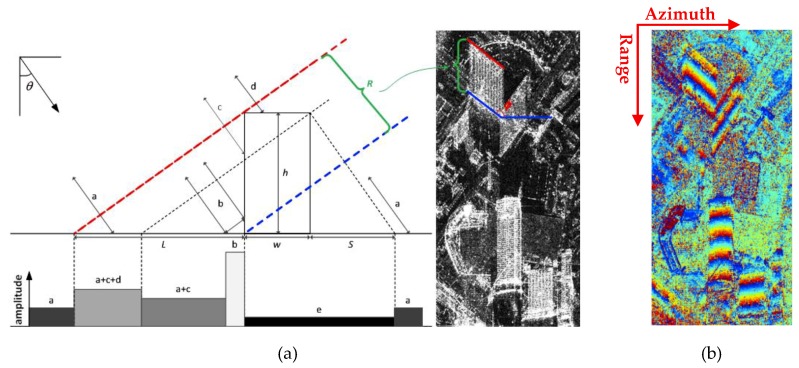
(**a**) The scattering model and projection geometry of a high-rise building in side-looking imaging; (**b**) the interferometric wrapped phase of high-rise building layover area.

**Figure 2 sensors-19-01439-f002:**
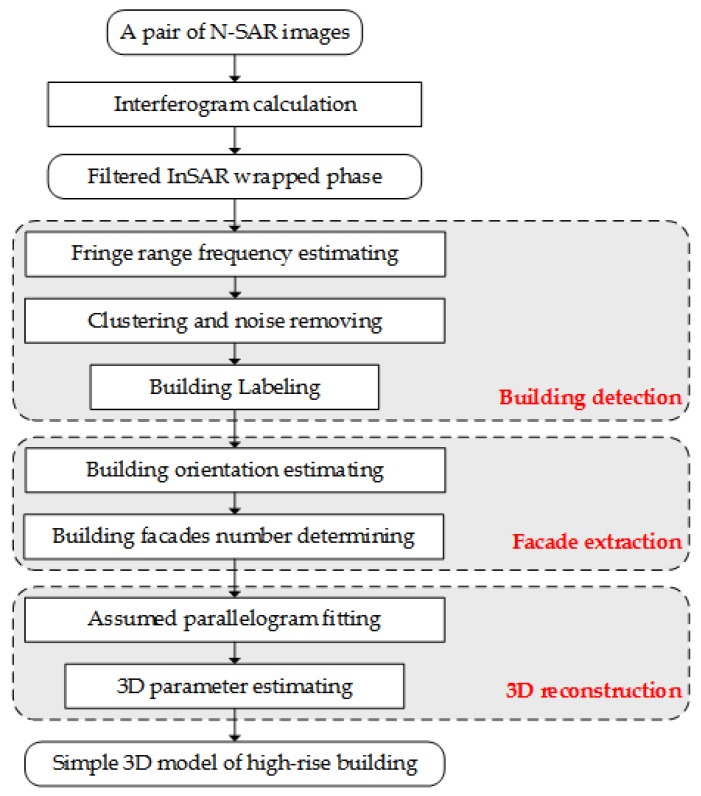
Workflow of the proposed approach.

**Figure 3 sensors-19-01439-f003:**
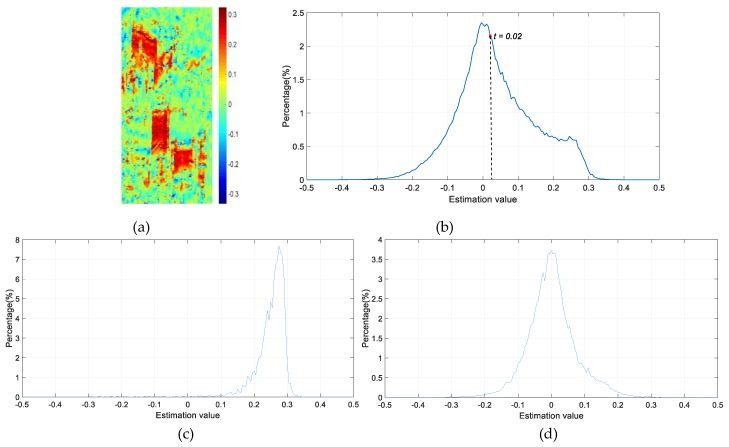
(**a**) The value of the local frequency estimation result of Figure 2b. (**b**) The statistical distribution of these values in Figure 4a. (**c)** Estimation values distribution in layover area. (**d**) Estimation values distribution in non-layover area.

**Figure 4 sensors-19-01439-f004:**
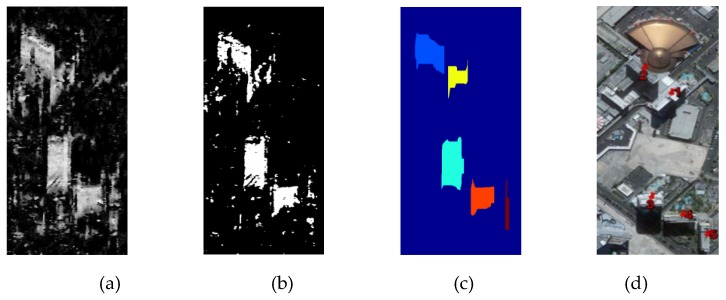
(**a**) Layover candidates after thresholding. (**b**) Initial layover map after *k-means* classification. (**c**) Final labelled building layover patches. (**d**) Google Earth optical image.

**Figure 5 sensors-19-01439-f005:**
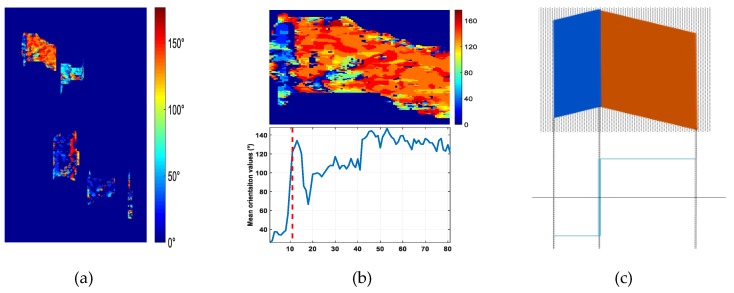
(**a**) Orientation map. (**b**) Zoomed-in image of Building 2. (**c**) Determination of number of visible façade orientation analysis.

**Figure 6 sensors-19-01439-f006:**
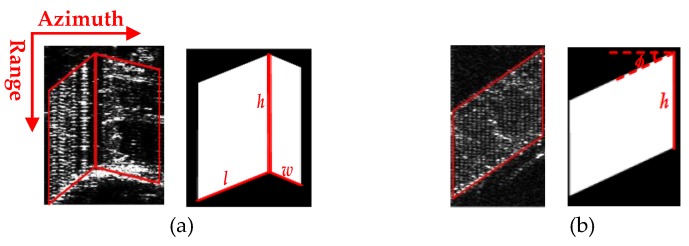
SAR projection of single high-rise building layover and corresponding parallogram shapes (**a**) with two visible façades (**b**) only one visible façade.

**Figure 7 sensors-19-01439-f007:**
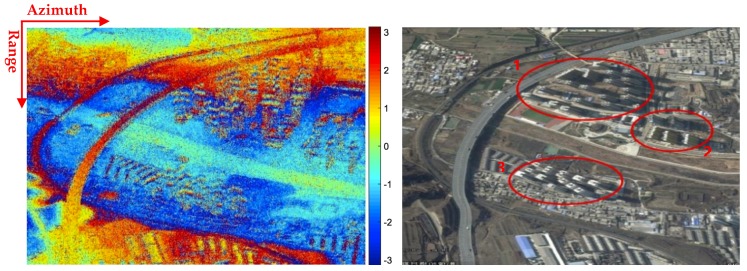
(**a**) Interferometric SAR (InSAR) phase imagery from airborne N-SAR system; (**b**) optical image (@ Google Earth) with marked high-rise building areas.

**Figure 8 sensors-19-01439-f008:**
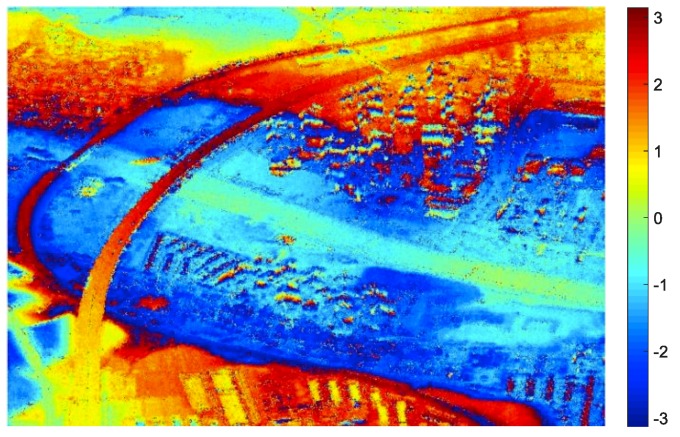
InSAR phase after nonlocal filtering.

**Figure 9 sensors-19-01439-f009:**
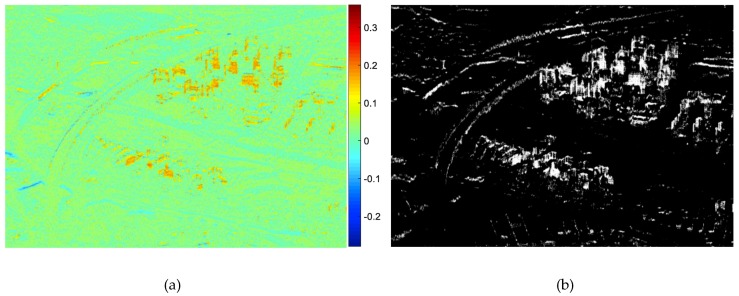
(**a**) Local frequency estimation result; (**b**) layover candidates after thresholding; (**c**) *k*-means filtering result; (**d**) final labelled layovers; (**e**) orientation image.

**Figure 10 sensors-19-01439-f010:**
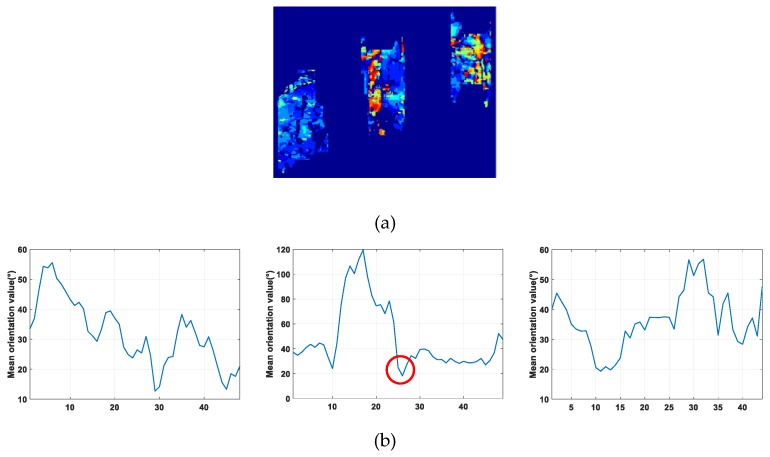
(**a**) Enlarged orientation map of Building 1-3 in Area 3; (**b**) mean orientation values along column of Building 1-3.

**Figure 11 sensors-19-01439-f011:**
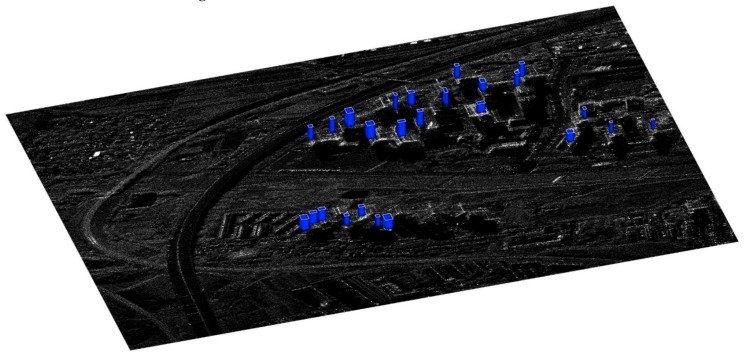
Reconstructed three-dimensional model of high-rise buildings overlaid on synthetic aperture radar (SAR) amplitude imagery.

**Table 1 sensors-19-01439-t001:** Retrieved three-dimensional (3D) parameters of these high-rise buildings in the scene.

	Building No.	Detected or Not	Façade number	Length	Width	Height	Mean height and error
**Area 1 with 18 high-rise buildings** (Detected: 14/18 = 77.8%; 2- façades extracted: 6/14 = 42.9%)	1 2 3 4 5 6 7 8 9 10 11 12 13 14 15 16 17 18	Yes Yes Yes Yes No No Yes Yes Yes Yes No Yes Yes No Yes Yes Yes Yes	1 1 2 2 - - 2 2 1 1 - 2 1 - 1 2 1 1	15.52 m 22.68 m 25.56 m 27.36 m - - 26.28 m 10.80 m 19.44 m 21.60 m - 11.89 m 27.72 m - 15.48 m 19.12 m 15.12 m 16.92m	- - 17.28 m 13.68 m - - 7.92 m 8.30 m - - - 16.56 m - - - 13.26 m - -	36.28 m 39.59 m 38.71 m 38.37 m - - 40.95 m 40.69 m 40.46 m 34.31 m - 36.51 m 25.29 m - 34.53 m 24.39 m 36.95 m 36.73m	35.98 m (1.98m)
**Area 2 with 5 high-rise buildings** (Detected: 4/5 = 80%; 2- façades extracted s: 3/4 = 75%)	1 2 3 4 5	Yes Yes No Yes Yes	2 2 - 2 1	21.96 m 10.50 m - 10.92 m 10.58m	12.24 m 8.95 m - 5.58 m 3.46m	24.85 m 25.73 m - 29.91 m 27.05m	26.89 m (−7.11m)
**Area 3 with 12 high-rise buildings** (Detected: 7/12 = 58.3%; 2- façades extracted: 2/7 = 28.6%)	1 2 3 4 5 6 7 8 9 10 11 12	Yes Yes Yes No No No Yes No Yes Yes No Yes	1 2 1 - - - 1 - 1 1 - 2	25.83 m 36.02 m 24.95 m - - - 16.65 m - 20.16 m 14.36 m - 27.41m	- 15.48 m - - - - - - - - - 19.44m	31.23 m 31.67 m 31.90 m - - - 27.05 m - 26.40 m 27.05 m - 25.17m	28.64 m (−5.36m)

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
