# Peer review of "High-Rise Building 3D Reconstruction with the Wrapped Interferometric Phase"

_sensors, 2019, doi:10.3390/s19061439_

Reviewer 1 Report

The authors had made extensive revisions and emphasized their novelty.

Author Response

Thanks for the reviewer’s former comments to help us improve the manuscript. 

Reviewer 2 Report

The revised manuscript has been improved according to the reviewers' comments and suggestions. However, some comments have not been answered sufficiently. The comments from 1 to 3 are related to the previous review. The others are related to the revised part in this round.

1.The authors' answer for the comment 4 of the reviewer 2 is not clear. Please mention clearly whether this manuscript considers the difference between the slant and grand range lengths in calculating length l and width w or not.

2.The authors' answer for the comment 7 of the reviewer 2 is not sufficient. Which angle faces to the range direction? Does it correspond to θ in Figure 6(b)? Please mention the definition in the manuscript with the direction (clockwise or counter-clockwise). Figure 5(b) in the revised manuscript is easy to visually recognize two facades. However, the reviewer cannot still recognize which buildings in Figure 9(e) have two facades even when this figure is enlarged by using a PDF reader software. For example, it is suggested to show one more panel in which the region around the buildings 1-3 in Area 3 is enlarged and to discuss how the proposed method recognized that the building 1 has two facades but the others have only one facade. The plot as shown in the lower panel in Figure 5(b) for each building would be helpful to demonstrate the validity of the proposed method.

3.The authors' answer for the comment 8 of the reviewer 2 is not sufficient. At first, Figure 13 is drawn with a different style from that of Figures 7-9, and it is hard to compare them. To present the results in Table 1 clearly, the reviewer suggests indicating the building number in Figure 9(d) or 9(e). Next, please mention the meaning of the number with circle in Figure 13. Is the position of No. 3 in Area 2 correct?  The reviewer thinks it should be above No.4 by referencing the Figure 7(b). Please verify the positions of all buildings again. Due to the low contrast in Figure 13, it is very hard to recognize building layovers in this figure. 

4. In Figure 5(b). The reviewer suggests to align the horizontal size of upper and lower panels for easy recognition. Please mention what the orange vertical line in the lower panel is. There is a small orientation angle gap around the center, but this is not selected as the separator between facades. Please explain the threshold of the orientation angle gap to determine the number of facade if the determination is automatically done in the proposed method.   

5. In Figure 4(c). The building 5 looks clearly divided in upper and lower parts in Figure 4(b) but the two parts are unified in Figure 4(c). It implies that the morphological process acts very strongly. Please describe in the manuscript the manually selected parameters of opening/closing for Figure 4 in both range and azimuth directions, respectively. Do these manually selected parameters differ from those in Figure 7? In Line 201, the authors mention that these parameters depend on the datasets. On the other hand, in Line 319, the authors mention that the parameters used in Section 3 are fixed. This is confusing.

Author Response

Thanks very much for the reviewer’s excellent suggestions before and now. These suggestions help us make the manuscript much better.

The attached file include the detailed reply to all the comments.

Reviewer 3 Report

1. Most of the present tense should be changed to past tense in the paper, such as P1L18 “... urban scene is studied...”, P1L20 “... high-rise building are detected...”, etc.

2. The authors said “there is few researches focusing on the high-rise buildings in the wide urban scene among the aforementioned papers”. Why “wide urban scene” is important?

3. To highlight the value of this research work, it is strongly suggested that the authors briefly explain what are the key challenges in the process of building reconstruction with wrapped InSAR and why it is important to study on this method.

4. In the experiment part, I think the authors should give some explanation about the computing environment and the execution efficiency of the method.

5. No ground truth verification, and even the comparison with other method, is a deficiency of the paper.

Author Response

Thanks for the reviewer's comments and for taking the time to consider our paper.

The attached file includes the reply to the comments.

Round  2

Reviewer 2 Report

In this round, all of the major problems have been improved. The comments below are minor ones including the authors' simple mistakes. The number of comments is same as the previous ones of the reviewer 2.

1.The reviewer is satisfied because the answer is clear (this manuscript does not consider the difference), although the reviewer cannot agree with that the difference does not need to be considered.

 2.The answer is clear. However, the modifications of the manuscript are insufficient. The range direction and azimuth direction are not illustrated in Figure 6. Figure 9(f) is absent. In the added sentence in line 313-316, the latter part is missing. Please check again.

 3.Please add the explanation of the building number in Figure 9(d) in the manuscript for the benefit of readers.

 4.The answer is clear. Please mention in the manuscript the gap needs to be above 90 degrees to be selected as the separator. Otherwise, the readers cannot know the threshold for this selection.

 5. The reviewer recommends to modify the last part in Line 327 from "are all manually fixed constant" to "are all manually fixed constant within the whole scene in Figure 7".

Author Response

Thanks for the excellent suggestions from the reviewer, and our reply to comments is in the PDF file.

Reviewer 3 Report

1. The paper said “3D information of buildings were estimated”. The word “estimated” is not scientific. What about the precision? How meaningful the method is, when the results could only be estimated?

2. The authors should summarize more and highlight the significance and reference of their research work. It should not be just a method customized for 3D reconstruction of Chinese buildings in some practical project. It should have some scientific meanings.

3. All the functions should be listed separately from the text paragraphs, e.g. the functions in Section 2.1 and Section 3.2.

4. Experiment part was too simple. There was not discussion about the results. How to verify the experimental results? And comparing with other methods, what were the advantages of the proposed method?

Author Response

Thanks for the reviewer for take time to consider our paper. And our reply to the comments is in the PDF file.
